# The impact of a school garden program on children's food literacy, climate change literacy, school motivation, and physical activity: A study protocol

Anna Stage[1,2☯*], Marie Caroline Vermund[1,2,3☯], Mads Bølling[2,4], Camilla Roed Otte[3], Alberte Laura Oest Müllertz[2], Peter Bentsen[2,5], Glen Nielsen[1], Peter Elsborg[2]

1 Department of Nutrition, Exercise, and Sports, University of Copenhagen, Copenhagen, Denmark, 2 Center for Clinical Research and Prevention, Copenhagen University Hospital – Bispebjerg and Frederiksberg, Copenhagen, Denmark, 3 Haver til Maver, Copenhagen, Denmark, 4 VIA University College, Research Centre for Pedagogy and Bildung, Program on Outdoor Pedagogy, Aarhus, Denmark, 5 Department of Geoscience and Natural Resource Management, University of Copenhagen, Copenhagen, Denmark

☯ These authors contributed equally to this work.
* anna.stage.hansen@regionh.dk

## Abstract

### Objective

FoodACT aims to investigate how school gardens affect children's food literacy (FL), climate change literacy (CCL), school motivation (SM), and physical activity (PA).

### Design

It comprises a multimethod, quasi-experimental inquiry into an existing Danish school garden program, Gardens to Bellies (GtB). Data will be collected using surveys, accelerometry, semi-structured and focus-group interviews. The study is preregistered with ClinicalTrials.gov (#NCT05839080).

### Setting

Six GtB school garden locations across Region Zealand and Region of Southern Denmark.

### Participants

Fourth grade pupils attending GtB (approx. 1600) are recruited to the intervention group. Fourth grade pupils from schools not attending GtB (approx. 1600) are recruited to the control group.

### Intervention

Pupils grow, prepare and cook foods for meals in the school garden during eight garden sessions.

**Data availability statement:** No datasets were generated or analysed during the current study. All relevant data from this study will be made available upon study completion.

**Funding:** This work was supported by a research grant from the Novo Nordisk Foundation (Grant number: NNF22SHO077522) and the Innovation Fund Denmark (Grant number 3129-00058B). The funding sources were not involved in any part of this study and did not influence the decision to submit the paper for publication.

**Competing interests:** MCV and CRO are employed at Gardens to Bellies / Haver til Maver. This does not alter our adherence to PLOS ONE policies on sharing data and materials.

## Main outcome measures

FL, CCL and SM are measured using pre- and post-intervention surveys in both groups. Pupils participating in GtB have their PA assessed using accelerometery, and acute SM by text-message-surveys. Semi-structured and focus-groups interviews are held with garden facilitators and pupils focusing on the implementation of GtB and mechanisms related to developing FL and CCL.

## Analysis

The effect on FL, CCL and SM is assessed using linear mixed models. PA and acute SM are assessed by comparing data on days with and without GtB in a subsample of 900 pupils. Qualitative data will be analysed using thematic analysis.

## 1. Introduction

School gardens have emerged as a widespread initiative across educational institutions worldwide as they hold promise in enhancing children's overall health and well-being while fostering social development and academic performance [1–3]. School gardens can be situated within the widely used pedagogical approach called Education Outside the Classroom (EOtC), which can be broadly defined as learning experiences taking place in outdoor settings such as green spaces, cultural institutions, or companies as a supplement to classroom teaching [4]. Studies have shown that children participating in EOtC benefit in various aspects of school-related well-being [5,6]. Through the EOtC approach, children achieve higher levels of pro-social behavior [7], school motivation (SM) [8], and reading competencies [9–11]. Furthermore, EOtC has been associated with more physical activity (PA) [12,13]. These results may translate to the school garden as a specific type of EOtC [14–17].

A systematic review from 2016 by Ohly et al. concluded from qualitative findings that school gardens can affect pupils' well-being and motivation positively through the experience of achievement and personal satisfaction arising from the knowledge and practical skills acquired through seeding, growing, and harvesting crops. However, the review also concluded that more robust quantitative research is needed to support these findings [18]. On the contrary, a systematic review from 2017 by Savoie-Roskos et al. found that although quantitative evidence is mixed and fraught with limitations, most studies on the potential for garden-based interventions to improve fruit and vegetable intake among children and youth indicate small but positive influences [19]. This is complemented by findings in a recent systematic review from 2024 by Vaughan et al. which indicates that cooking programs for children can result in small improvements in cooking efficiency and vegetable intake [20]. Besides being a health and well-being promoting initiative, the educational setting of school gardens may provide unique hands-on opportunities for pupils to develop their knowledge of and skills to plan, select, and prepare healthy and environmentally sustainable food [1,18,21,22]. The acquisition of these competencies forms a crucial foundation for a wider array of sustainable behaviors, which in turn can help alleviate the adverse effects of climate change [23]. This includes the promotion of more sustainable food consumption, which is crucial in a time where the escalating global consumption of ultra-processed and animal-derived foods is acknowledged as the main catalyst for diet-related health issues and climate change [24]. While promoting knowledge and skills that can enable more environmentally sustainable food consumption, studies also suggest that school gardens foster high-intensity PA through bodily work such as digging, lifting, and watering the plots [25–27]. In other words, school gardens have the

potential to affect both sides of the energy balance equation in terms of dietary intake and PA [28]. Additionally, school gardens have the potential to enhance children's SM by providing a hands-on learning experience that encourages practical skills and collaboration [29]. This is particularly significant given the recent decline in well-being and motivation for school [30].

School garden programs thus hold a strong potential for supporting action towards the combined prevention of current public health challenges, such as poor dietary behaviors, sedentary lifestyles, declining SM, and climate change. However, more evidence is needed to support the potential benefits and to further understand how they are achieved through school gardens. Specifically, more evidence on the benefits of school gardens in a European and/or Scandinavian context is needed as the majority of existing evidence relies on studies from the U.S. and Australia [22]. Previous investigations of school garden programs have primarily focused on outcomes related to PA [1,18,22] and to specific food intake such as fruit and vegetable consumption [1,18,22] and metabolic health parameters[11]. Research in this field has thus relied on a narrow range of outcomes, and more robust quantitative research is needed focusing on a broader set of outcomes important to children's health and well-being to support the qualitative evidence suggesting a wide range of benefits on well-being, motivation and dietary intake [18].

Recent studies highlight food literacy (FL) and climate change literacy (CCL) as potential predictors of dietary and sustainable behaviours [31,32]. The aim of interventions targeting these literacies is to foster a wide range of interrelated knowledge, skills, attitudes, and behaviours of importance to food and climate change [33–35]. FL and CCL is crucial for promoting sustainable dietary behaviors as food systems account for approx. one-quarter of global greenhouse gas emissions [24,36] whilst 30–40 pct. of food produced in the world is wasted [37]. Improving CCL can thus empower people to make more informed, sustainable decisions about their dietary behaviors over the life-course [37].

School gardens provide hands-on, experiential learning about environmental topics like sustainable agriculture, composting, organic produce, and the impact of agriculture on the environment. This helps children develop a stronger connection to nature, the food they eat, an increased understanding of climate change, and concrete skills necessary for engaging in actions to combat climate change [38]. School gardens can thus be seen as a way to foster children's relationships with nature and develop their sense of stewardship through developing CCL, which is an important factor in addressing climate change through individual behaviors and actions [39]. However, research on the potential for school gardens to promote such broad conceptualizations of FL and CCL remains scarce in the existing literature [18]. Since the majority of school garden studies have focussed on nutritional outcomes such as fruit and vegetable intake, there is a particular need for more research on its potential to promote children's FL [20]. As children's health and health behaviours are formed during childhood and adolescence, the combined promotion of FL, CCL, SM and PA in the school setting holds the potential for promoting health across the life-course [40–42]. However, there is a lack of knowledge on the underlying mechanisms for and impact of school garden programs aiming to promote FL, CCL, SM and PA simultaneously.

The aim of the FoodACT study is to investigate the impact and mechanisms of an already established school garden program on children's FL, CCL, SM, and PA. The Garden to Bellies (GtB) school garden program has been operational in Denmark since 2006. The program is chosen as a natural experiment intervention in the FoodACT study since it is already widely applied in Denmark and therefore feasible with potential for scalability in a real-world setting [43]. Identifying the impacts of interventions that are feasible, implementable, and scalable in real-world settings, such as GtB, will result in evidence that can have a high societal impact and relevance (i.e., ecological validity) [44]. However, when assessing the effects of a

school program that operates independently of the study, it becomes impractical to employ the attributes of a randomized controlled trial (RCT) design, such as randomizing pupils into intervention or control groups or exercising greater control over the activities of both implementers and pupils. Despite the challenges posed by this complexity, and the need to eliminate biases through an appropriate study design, the evaluation of real-world natural experiment interventions offers a distinct advantage in terms of opportunistic evidence, flexibility, and investigation of delivery at larger scales [45]. The purpose of this paper is therefore to present and discuss the hypotheses, study designs, data collection procedures, measures, and analytical strategies, that will be applied in the FoodACT study.

The study will investigate two hypotheses which are supplemented by a process evaluation of implementation and mechanisms. The testing of the two hypotheses and the process evaluation will be investigated in three sub-studies, as illustrated in Figs 1 and 2 and described in Table 1.

## 2. Materials and methods

The FoodACT study will comprise a multimethod, quasi-experimental inquiry into the existing school garden program GtB. RCTs are widely used in educational research to reduce bias in estimating the effects of the intervention [46]. RCT studies have previously shown health effects of school garden interventions [11]. However, when applied to school settings, RCTs necessitate a somewhat artificial ("unnatural") intervention from outside the school system itself, which does not reflect how educational initiatives or programs are implemented, established, and scaled up in real life. In other words, RCT designs enable high

| | Enrolment | Pre-intervention A | Pre-intervention B | Intervention start | Post-intervention A |
|---|---|---|---|---|---|
| Time point | -6 mo. | -3 mo. | -1 mo. | 0 mo. | 1 mo. |
| **ENROLMENT** | | | | | |
| Recruiting | x | | | | |
| Informed consent | | x | | | |
| **INTERVENTION** | | | | | |
| Garden to Bellies | | | | x | |
| **ASSESSMENTS** | | | | | |
| *Children* | | | | | |
| PA behaviours (device based, 2x7days) | | | | x | |
| Acute school motivation (text-message survey) | | | | x | |
| FL, CCL & SM (survey) | | | x | | x |
| Intervention experience (interview) | | | | | x |
| *Garden facilitators* | | | | | |
| Interview | | | | x | |

**Fig 1. Participant allocation timeline according to SPIRIT.**

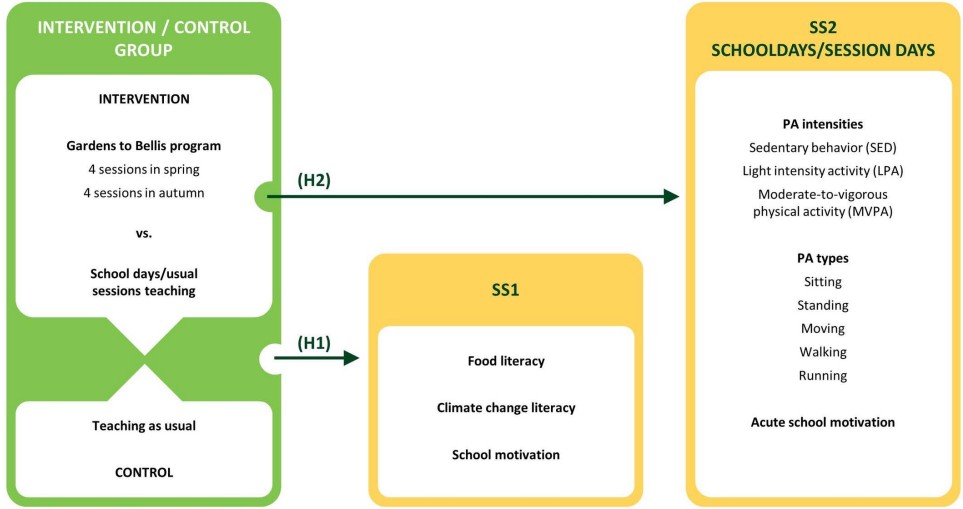

**Fig 2. Illustration of the two study hypothesis.**

**Table 1. Hypothesis and secondary aim of the FoodACT protocol.**

| Hypothesis 1 | Pupils participating in the school garden program develop higher levels of FL, CCL, and SM compared to pupils that undergo teaching as usual. | Sub-study 1 (SS1) |
|---|---|---|
| Hypothesis 2 | The pupils' PA levels are higher and involve different patterns of movement on school garden days compared to usual school days. | Sub-study 2 (SS2) |
| Process evaluation research question | Was the GtB program implemented as intended, and how does underlying mechanisms operating in the school gardens relate to the potential effects on FL and CCL? | Sub-study 3 (SS3) |

internal validity regarding effects but often have low external pragmatic/ecological validity because the intervention investigated does not reflect real-life school practices well and often cannot be implemented long-term or scaled up in municipals and schools without significant alterations. The aim of the FoodAct study was to study the impacts of an already established school-garden program, namely the GtB program. This was chosen in order to ensure scalability and sustainable implementation of the intervention studied which is important for pragmatic validity or practical/political relevance of the study. The GtB program is implemented at the municipality level, which means that it is a political decision to financially prioritize whether the schools within the municipality get to participate in GtB or not. Randomizing which schools or classes would get to participate is thus not acceptable from a municipal perspective (in Denmark) making it impossible to recruit municipalities and hence schools to an RCT trial of this school garden program. We therefore chose a natural experiment approach [47–49] to evaluate the outcomes associated with participating in an already developed, established, and widely used school garden program in Denmark. Based on the practical nature of the GtB program, the study will be a non-randomized controlled natural experiment [44], drawing on various study designs through sub-studies (see Fig 3), which will be elaborated upon in the following sections.

## 2.1 The Gardens to Bellies program

The Danish school garden program GtB will serve as an intervention in this study (see the FoodACT study Logic Model, Fig 4). The GtB program is an ongoing educational initiative

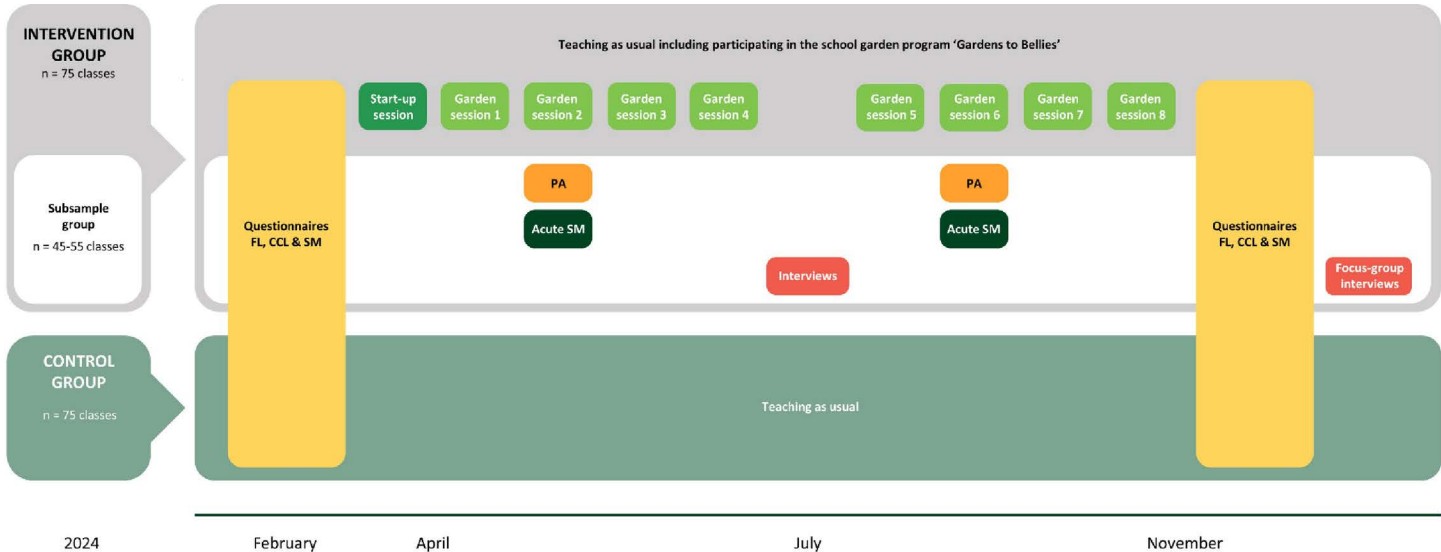

**Fig 3. Study design and timeline for the FoodACT study.**

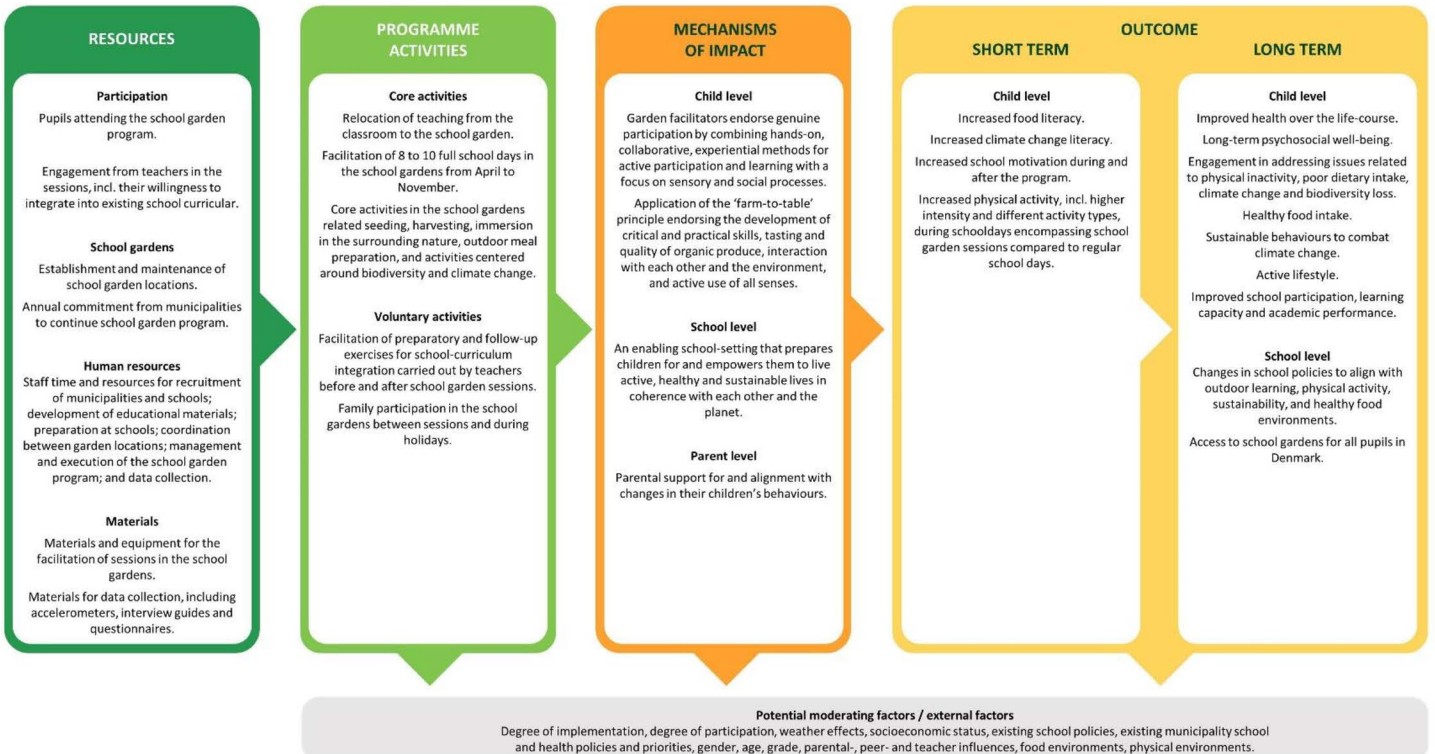

**Fig 4. The FoodACT Logic Model.**

launched in 2006. School classes are invited to attend eight sessions (whole school days) in the school gardens (ten sessions/days in a few select municipalities) over the course of the second semester of fourth grade and the first semester of fifth grade (pupils approx. ten years of age, range nine to twelve). The school garden sessions start each year in March and end in

October. Before the first school garden visit, an introductory workshop of one to two hours is conducted at the schools to spike the pupils' curiosity. The school gardens are currently located in nine municipalities across Denmark. All participating school classes attend a school garden located in their own or near their own municipality, hence, the distance to the school garden as well as their mode of transportation to and from the school garden may vary across the participating schools. Furthermore, the plot size of the gardens varies across the garden locations.

Activities in the gardens are led by garden facilitators who are trained in the culinary arts, biology, or gardening. Pupils are divided into smaller groups that are assigned a plot of land, which they are responsible for preparing, weeding, seeding, and harvesting. The overarching purpose is for pupils to learn where their food and produce comes from, how to take care of crops, and how to prepare meals with the vegetables and fruits they grow in the garden or find in the surrounding nature. The activities relate not only to the practical aspects of growing and preparing foods, but also to incorporating more theoretical elements related to climate change, biodiversity, and organic approaches to food production. The teachers are encouraged to support the garden facilitators during the sessions and to integrate the themes and learnings in the gardens into their regular school curriculum. During school holidays, pupils and their families are encouraged to water and weed the vegetables in the school gardens by themselves.

The activities of GtB are rooted in a facilitative approach through which the pupils are actively engaged. Emphasis is put on the development of critical and practical skills, tasting and quality of organic and seasonal produce, interactions with each other and the environment, and active use of all senses. Activities build on genuine participation by combining hands-on, experiential methods for active participation with a focus on sensory and social processes [48]. The program draws upon several theories and concepts related to outdoor learning. These include Place-Based Learning by using the local environment as a primary educational resource, fostering a deeper connection between students and their community. Furthermore, Problem-Based Learning engages pupils in solving real-world problems, enhancing critical thinking and collaborative skills. Embodied Learning highlights the importance of physical interaction with the environment, allowing students to learn through movement and sensory experiences, leading to a more profound and retained understanding. Cooperative Learning involves pupils working in groups to achieve common goals, promoting teamwork and social skills. Additionally, Experiential Learning, which focuses on learning through direct experience, and Inquiry-Based Learning, which encourages pupils to ask questions and explore topics in depth, can also play significant roles. These approaches collectively enhance learning in the outdoors and external learning environments by making education more interactive, relevant, and deeply connected to the pupil's physical and social environments [4]. A TIDieR checklist [50] and a detailed overview of the school garden program can be found in S1 File.

## 2.2  Recruitment and participants

The recruitment and enrollment of fourth grade pupils in the study was first initiated in September 2023 and ended in January 2024. Municipalities decide whether to offer the GtB program to schools within their district, which then sign up for the program on a yearly basis. All contact information for the schools will be provided by GtB. All contact information for the schools will be provided by GtB. School classes from public primary (non-private) schools that are signed up for the GtB program in 2024 but will not yet have started the program will be invited to participate in the intervention group via e-mail. Similarly, school classes who have not previously been and will not be enrolled in GtB in 2024 will be invited to participate in the control group via e-mail. The recruitment and inclusion of control classes will be based

on similarity with the intervention schools regarding geographical proximity and average disposable income at the municipal level to enhance comparability between the intervention and control schools. First, a list will be comprised of municipalities that approximately match intervention municipalities on average disposable income (based on 2017 data from Statistics Denmark) [51] and are in the closest possible geographic proximity to the intervention municipality. For example, intervention schools located on the Danish Island of Funen will be matched with control schools in a different municipality on Funen but with a similar municipal average disposable income. Based on this list, public primary (non-private) schools within suitable control municipalities will be invited to participate, and an even distribution of school classes within the matched municipalities will be attempted. However, all schools indicating that they wish to participate will be enrolled in the study.

As part of the recruitment, teachers will be asked to provide all parents with written information about the study. Parents will subsequently be invited to provide electronic informed consent for their children's participation. For SS2 and SS3, pupils will be asked to provide oral consent on the day of data collection. The recruitment process will be supported by general information available on the study project website [52]. By the end of data collection, all participating classes, including the control group, will receive a reimbursement equivalent to 200 EUR in recognition of their contribution.

The study will strive to include all schools participating in GtB in 2024 in the study's intervention group with no upper limit for participation. Sampling for the control group will aim to match the number school classes in the intervention group. For SS1, the aim will be to recruit 75 school classes (approx. 1600 pupils) for the intervention group and 75 classes for the control group. SS2 will be based on a subsample of 45–55 school classes from the intervention group, whereas SS3 will be based on a subsample of 20–25 pupils from the intervention group. Please, see Fig 2 for the main components and timeline of the study.

## 2.3  Research ethics

Danish law requires only research projects of biomedical character or studies involving risks to participants to have its ethics reviewed by a Regional Ethics Board; all other research projects are exempt from applying for formal ethical approval. The FoodACT study protocol was evaluated by the Regional Committee on Research Ethics and was deemed exempt of any ethics approval (protocol no: H-23010743). Parents/guardians of the pupils will be asked to provide an informed written consent for participation in the study. They will be informed about the opportunity to withdraw from the study at any time. Participants and schools will be anonymized, and a personal ID number will be provided to each participant prior to the data collection and subsequent analyses. The study is preregistered with ClinicalTrials.gov (#NCT05839080) [52]. All data and personal information will be handled in accordance with the Danish Data Protection Act. Data were not accessed before the submission of this protocol. Trial data documents will be available upon request.

## 2.4  Sub-study 1: Effects of GtB on food literacy, climate change literacy, and school motivation

**2.4.1  Design.**  SS1 will be designed as a clustered quasi-experimental study. The pupils' FL, CCL, and SM will be measured with online questionnaires pre- and post-program in both the intervention and control group. The completion of questionnaires will be facilitated by the teacher responsible for the class's participation in the study without the presence of the research team. Baseline data collection will commence one week prior to the start of the GtB program in the intervention group and within a matched two-week timeframe for the control

group. Follow-up data collection will commence one week after the end of the GtB program in the intervention group, and accordingly, within a matched two-week timeframe for the control group. All questionnaires are in Danish and will be administered during school hours on computers with brief videos introducing each questionnaire.

**2.4.2 Outcome measures.** *Food literacy (FL):* FL will be measured using the Food Literacy Questionary designed for school children (FLQ-sc), which is a 37-item questionnaire used to assess individual FL based on Benn's (2014) five FL competencies: 'to know', 'to do', 'to sense', 'to care', and 'to want' [32]. The questionnaire was developed and validated in a Danish context for pupils in sixth and seventh grade (aged 12–14) and was slightly modified to suit the age group in this study (aged nine to 12). This included the exclusion of item nine regarding experimenting with recipes as well as minor alterations to the wording of item 23 regarding the food pyramid. The FLQ-sc used in this study will thus contain 36 instead of 37 items. Items will be answered on a Likert scale. Five of these items assess frequencies, ranging from zero (never) to three (always). Thirteen items evaluated the degree of difficulty, from zero (very difficult) to three (very easy). Six items assessed abilities, from zero (to a very low degree) to three (to a very high degree). Lastly, 11 items evaluated opinions, from zero (strongly disagree) to three (strongly agree). Furthermore, two items in the "to know" scale are scored based on a count (zero to three) of how many correct answers the participants provided for three true/false questions (example of item: "Which food items contain high levels of protein? (Mark with a cross in each line.") The overall FL score will be calculated as the average of all items and used as a continuous variable ranging from one (lowest) to four (highest) in the analyses. [34].

*Climate change literacy (CCL):* An instrument for measuring CCL in pupils aged nine to 12 will be developed for the purpose of this study, since an age-appropriate and validated measure to assess CCL in children aged nine to 12 is yet to be devised [33,35]. The new instrument will be based on existing theoretical frameworks and related instruments such as the Climate Change Attitude Survey [45] and the Climate Capability Scale [31]. The development of the questionnaire will be guided by Boateng et al.'s (2018) 'Best Practices for Developing and Validating Scales for Health, Social, and Behavioral Research: A primer' [53]. The questionnaire will consist of items related to knowledge about climate change, practical skills and behaviors, attitudes and motivation, and the connection between individual actions and societal impacts. Items will be answered on a Likert scale with scores ranging from one to four. An overall CCL score will be measured as the average of all items on a score from one (lowest) to four (highest) and applied as a continuous variable in the analyses.

*School motivation (SM):* The Academic Self-Regulation Questionnaire (SRQ-A) [54] will be used to measure SM. SRQ-A is based on the Self-Determination Theory [54] and is developed for measuring the degree of autonomy exhibited by students across various academic tasks. The original questionnaire consists of 32 items with a four-point response scale that measures pupils' reasons for doing homework, classwork, trying to answer questions in class and trying to do well in school. Responses to items are scored on a four-point Likert scale (one= Not at all true, two= Not very true, three= Sort of true, four = Very true). However, as the application of homework varies in fourth and fifth grade in Danish schools, the items for this domain will be omitted reducing the scale to 24 items with seven items measuring the level of external regulation, seven items measuring introjected regulation, five items measuring identified regulation and five items measuring intrinsic motivation.

**2.4.3 Background descriptive measures.** The inclusion of confounders described in this section are based on previous literature [1,18,26,27]. Socio-economic status (SES), and gender will be used in all analyses. Age will be used in analyses related to FL and CCL. Food and cooking class participation will be included in analyses related to FL. Other covariates, such as

related school activities, transport, weather, and the size of the school garden will be included in analyses relevant to PA.

***Socio-economic status (SES):*** When providing informed consent on behalf of their child(ren), information on parental educational background and occupation will be collected. Parents' or guardians' social class will be categorized using the Danish Occupational Social Class coding-scheme [55], which categorizes social class from one (highest) to six (lowest) based on, first, occupation, and, in case the occupation is not provided, educational background. The pupils' SES will then be based on the parent or guardian with the highest social class in the case of joint custody. In the case of full custody, the pupils' SES will be based on the social class of only this parent or guardian.

***Gender and age:*** Information on the pupils' gender identification and date of birth will be collected electronically from the parent or guardian concurrent with the provision of informed consent. Gender will be collected by asking whether the pupil identifies as 'girl', 'boy' or 'other' and is subsequently used categorically. Age will be calculated in years from the date of birth using first of March 2024 as an endpoint and included as a continuous variable in the analyses.

***Related school activities:*** Teachers responsible for the participation of each school class will be asked to provide class-level information on the pupils' school schedules. The information will be used assess whether the involved classes' participate in food and cooking class (also referred to as food science or home economics in other countries) as part of their school curriculum during fourth grade or the first semester of fifth grade (binary variable: yes/no).

**2.4.4 Sample size.** Based on the recruitment target of previous similar studies in a Danish school setting, power calculations were conducted using 25 school classes with an average of 18 pupils per class in each group to calculate a minimum detectable effect size. This approach was chosen due to the limited existing research on the effects of school gardens on the chosen outcomes of this study. Power calculations were completed in the R package PowerUpR. In the calculations, the clustered nature of data with pupils nested in classes was considered, conventional levels of statistical power (0.8), and 2-tailed level significance (0.05) were applied. An intraclass correlation (ICC) of 0.046 was used based on a similar dataset of pupils from a Danish study on EOtC [56]. Adjustments for three covariates were also included in the calculation. Power calculations show that with this sample, the minimum detectable effect size of Cohen's d would be 0.255, 95% CI [0.076,0.434]. This effect size is substantially smaller compared to what has been found in another less intensive school-based intervention aiming to increase pupils' FL [57]. This suggests that even though the intended recruitment level is not achieved, or a high drop-out rate becomes prevalent, the study will be sufficiently powered.

**2.4.5 Statistical analyses.** Linear mixed models will be applied to assess the intervention effects on FL, CCL, and SM [58]. Two-tailed tests with p-values below 0.05 will be considered statistically significant.

The analyses will account for the pupils' nesting in their school classes by introducing school class as a random effect in the models. Intervention effects will be estimated through the inclusion of a group variable (intervention or control) and by adjusting for the outcome measure at baseline as well as confounders. Significantly better developments in these measures in the intervention group compared to the control group when adjusting for baseline level and confounders will be interpreted as a positive effect of the GtB program. Due to the lack of randomization, the analyses will be adjusted for known confounders such as gender, and SES. The analysis on the effect of GtB on FL will be further adjusted for participation in food and cooking classes during fourth grade or first semester of fifth grade. The analyses on

the effect of GtB on FL and CCL will also be adjusted for age. Lastly, ICCs will be calculated by running the linear mixed models without the covariates to assess potential variation among school classes as fractions of the total variance. All analyses will be conducted using RStudio (version 1.4.1106; RStudio, Inc.) and IBM SPSS Statistics (version 29.0.1.0; IBM Corp.).

The models are expressed as follows:

***Effect of GtB on FL :*** $\text{Endline FL}(Y) = (\beta_0 + b_{class}) + \beta_1 \text{ FL baseline} + \beta_2 * \text{int\_con} + \beta_3 * \text{gender} + \beta_4 * \text{SES} + \beta_5 * \text{age} + \beta_6 \text{ food\_cooking\_class} + \varepsilon$

***Effect of GtB on CCL :*** $\text{Endline CCL}(Y) = (\beta_0 + b_{class}) + \beta_1 \text{ CCL baseline} + \beta_2 * \text{int\_con} + \beta_3 * \text{gender} + \beta_4 * \text{SES} + \beta_5 * \text{age} + \varepsilon$

***Effect of GtB on SM :*** $\text{Endline SM}(Y) = (\beta_0 + b_{class}) + \beta_1 \text{ SM baseline} + \beta_2 * \text{int\_con} + \beta_3 * \text{gender} + \beta_4 * \text{SES} + \varepsilon$

Explanation for model concept for effects on FL, CCL and SM:

1. Y = The dependent variable (outcome) being analyzed, in terms of a score after the intervention.

2. $\beta_0 + b_{class}$ = The intercept of the model.

3. $\beta_0$ represents the fixed intercept, which is the expected value of Y when all predictor variables are set to zero.

4. $B_{class}$ is the random intercept for class, which accounts for potential similarities among pupils within the same class.

5. $\beta_1$ * baseline = The effect of baseline for the outcome.

6. This controls for pupils' baseline levels before the intervention.

7. $\beta_1$ represents how changes in baseline influence Y.

8. $\beta_2$* int_con = The effect of *int_con*, represented by an intervention-control group variable.

9. $\beta_2$ captures the difference in Y (the outcome variable) between intervention and control groups.

10. $\beta_3$ * gender = The effect of gender.

11. Gender is coded as a categorical variable (0 = boy, 1 = girl, 2 = other). $\beta_3$ represents the difference in Y (outcome) between genders (boy, girl or other).

12. $\beta_4$* SES = The effect of socio-economic status (SES).

13. $\beta_4$ takes into account how changes in SES impact Y (the outcome).

14. $\beta_5$*age = The effect of age.

15. $\beta_5$ takes into account how changes in age impact Y (the outcome).

16. $\beta_{6*}$ food_cooking_class = The effect of food and cooking class.

17. $\beta_6$ takes into account how changes in food and cooking class impact Y (only for FL).

18. $\varepsilon$ = The residual error term, capturing unexplained variation in Y.

## 2.5 Sub-study 2: Physical activity and movement behavior in the school gardens

**2.5.1 Design.** A within-subject design will be applied for investigating whether, and to what extent, the pupils' PA and acute SM are different in the school gardens compared to teaching as usual (i.e., classroom-based teaching). PA will be measured using accelerometry for two weeks out of which the pupils will attend the school garden one day. The pupils' experience of basic psychological needs satisfaction, intrinsic motivation, and perception of the learning environment on school garden days and days with teaching as usual will be measured using an electronic survey distributed by a text-message to the parents on days they attend the school garden and on usual schooldays, in total seven days simultaneously with measuring PA.

**2.5.2 Outcome measures.** *Physical activity (PA):* Will be measured with SENS motion® accelerometers (SENS motion PULS pnr.A01.6, SENS Innovation, Copenhagen Denmark, 2024) [59]. The primary outcomes will be pupils' PA levels, i.e., sedentary behaviour (SED), light-physical-activity (LPA), moderate-physical-activity (MPA), and moderate-to-vigorous PA (MVPA), as well as activity types (lying, sitting, standing, walking, cycling, and walking) measured in minutes per day. The data will be included in the analysis as scale data with information on minutes spent in the PA level categories and in the categories of activity types.

*Psychological needs satisfaction and intrinsic motivation during schooldays with and without school garden sessions:* This will be measured using a 12-item instrument which measures the extent to which children experience intrinsic motivation/enjoyment (three items) and have their basic psychological needs for experiencing competence (three items), relatedness (three items), and autonomy (three items) satisfied as described by Self-Determination Theory and Basic Psychological Need Theory [54]. Responses to items are scored on a four-point Likert scale (one= Not at all true, two = Not very true, three= Sort of true, four = Very true) and will be included in the analysis as categorical data. The instrument has been developed for and pilot-tested in two previous school-based intervention studies [60,61]. The items for measuring experiences of intrinsic motivation, competence, and autonomy are based on the Interest/Enjoyment, Perceived Choice/Autonomy, and Perceived Competence sub-scales of the Intrinsic Motivation Inventory (IMI) [62]. The items for measuring experiences of relatedness are inspired by the relatedness satisfaction subscales of Basic Psychological Need Satisfaction Scales, Basic Psychological Need Satisfaction and Frustration Scale (BPNSNF) [63] and the Psychological Need States in Sport Scale (PNSS) [64].

*Learning environment perception:* Will be measured with the Learning Rating Scale (LRS). It is a four-item scale answering on a 10 centimetre visual analogue scale and evaluates the pupils' view of how much they learn at school, how well they are getting along, and whether the teaching methods suit them [65]. The 10 cm visual analogue scale ranges from zero to 100, not visible to the participant. The place where the participant chooses to put their answer provides a number from zero to 100, e.g., 46. The data will be included in the analysis as scale data.

**2.5.3 Covariates for SS2.** *Physical education (PE):* Information on PE (binary variable: yes/no on a daily basis) during their participation in the study was collected though school timetables.

*Transportation to/from the school garden:* Information on which transport mode; walking, biking, train, bus, or a combination, will be obtained from the responsible teachers, and used as a categorical variable.

*Weather data:* Data on general type of weather conditions, i.e., sun, rain, wind, thunder or snow, during school garden days will be collected from the Danish Metrologic Institute [66], and included as a categorical variable

***Size of the school garden plot:*** The plot size of the various school gardens will be measured in km² through the Geographic Information System and included as a scale variable [67].

**2.5.4 Sample size.** In SS2, a subsample of the intervention group containing 45–55 classes will be recruited based on the former estimate calculation described in section 2.4.4 This number of classes was chosen in terms of resources, practicality, and distance to the school gardens. However, school gardens from both rural and urban areas will be included to secure representativity in the sub-sample. Based on previous similar Danish school intervention studies, the estimated drop-out during the intervention is 7 pct. of classes, corresponding to four classes. Including the minimum of 45 classes with an estimated average of 18 pupils per class will provide 11.340 days with PA data from accelerometer based on 810 children since data is based on seven days with two measurements. A previous study has found statistically significant positive associations between EOtC-sessions and PA on an even smaller sample adopting the same design [12], which indicates that a sample of 18 pupils per class will be sufficient.

**2.5.5 Statistical analysis.** All analysis will be conducted in the statistical program RStudio (version 1.4.1106; RStudio, Inc.) [68]. Linear mixed models taking into account the clustered nature of data with days clustered in pupils clustered in classes will be conducted to test differences between the pupils' PA, basic psychological needs satisfaction, and intrinsic motivation on days with the school garden program and usual school days. Two-tailed tests with p-values below 0.05 will be considered statistically significant. All analyses will be adjusted for gender, and SES. Additionally, analyses adjusted for physical education, size of school garden and transport to/from the school garden will be applied in relevant sub-analysis.

The models are expressed as follows:

Acute SM :

$$\text{Mean acute SM}(Y) = (\beta_0 + b_{\text{class}} + b_{\text{id}}) + \beta_1 * \text{school\_schoolgarden} + \beta_2 * \text{gender} + \beta_3 * \text{SES} + \varepsilon$$

Acute PA :

$$\text{Mean PA}(Y) = (\beta_0 + b_{\text{class}} + b_{\text{id}}) + \beta_1 * \text{school\_schoolgarden} + \beta_2 * \text{gender} + \beta_3 * \text{SES} + \beta_4 * \text{weather} + \varepsilon$$

Explanation for model concept for effects on acute SM and PA:

1. Y = The dependent variable (outcome) being analyzed, such as a score after the intervention.

2. $\beta_0 + b_{\text{class}} + b_{\text{id}}$ = The intercept of the model.

3. $\beta_0$ represents the fixed intercept, which is the expected value of Y when all predictor variables are set to zero.

4. $\beta_{\text{class}}$ is the random intercept for class, which accounts for potential similarities among pupils within the same class.

5. $\beta_{\text{id}}$ is random intercept for individuals within each class, capturing within-class variation.

6. $\beta_1$ * school\_schoolgarden = The effect of being in a school garden versus a regular school day.

7. $\beta_1$ represents the difference in Y between school garden days and school days.

8. $\beta_2$ * gender = The effect of gender.

9. Gender is coded as a categorical variable (0 = boy, 1 = girl, 2 = other), $\beta_3$ represents the difference in Y (outcome) between genders (boy, girl or other).

10. $\beta_{3*}$ SES = The effect of socioeconomic status (SES).

11. $\beta_3$ shows how changes in SES impact Y.

12. $\beta_{4*}$ weather = the effect of weather.

13. $\beta_4$ shows how weather conditions influence Y.

14. $\varepsilon$ = The residual error term, capturing unexplained variation in Y.

## 2.6 Sub-study 3: Evaluation of implementation and mechanisms of impact

In SS3, the implementation and mechanisms of impact will be investigated by conducting a qualitative exploration of whether the GtB program activities have been implemented as intended, as well as how underlying mechanisms operating in the school gardens relate to the potential effects on FL and CCL [69]. SS3 will focus on how the GtB program induces change only in the outcomes of FL and CCL as these concepts, to the best of our knowledge, have not previously appeared in the school garden literature [18]. Comparatively, PA is a widely studied outcome in both school garden studies [1,18,22], and in EOtC studies overall, where SM is also a prevalent area of investigation [5,6]. The sub-study will follow the Medical Research Council's framework on developing and evaluating complex interventions [44] and standards for evaluation mechanisms [70], and will thus go beyond asking whether the GtB program is effective, to also unpack how it works, contributes to change, and interacts with the context in which it is implemented. Consequently, implementation will be understood as "*an actively planned and deliberately initiated effort with the intention to bring a given intervention into policy and practice within a particular setting*" [71](p6). By viewing the implementation of the GtB and its intended effects as inherently embedded within a specific context [71], SS3 will focus on identifying contextual factors important to the delivery of the program in its various unique settings. Mechanisms of impact will be operationalized as "*underlying entities, processes, or structures which operate in particular contexts to generate outcomes of interest*" [70](p368). By investigating GtB as an instrumental case, SS3 will further aim to identify significant mechanisms driving or hindering the promotion of FL and CCL in pupils. This will contribute with insights into how and why outcomes are generated, and whether certain contextual conditions affect these processes. Moreover, adherence to the intervention will be investigated by monitoring the dose of the intervention by asking the garden facilitators whether the frequency and duration of the garden sessions were delivered as intended [69].

**2.6.1 Participants and data collection.** A subset of garden facilitators and pupils from intervention schools will be selected purposefully to enable a diverse geographic and socio-economic representation. Two different qualitative methods will be applied to allow for triangulation of data, and thereby increase the validity and understanding of emerging themes [72]. Focus-group interviews will be held with 20–25 pupils from the intervention group distributed across four to five focus groups. As mechanisms of impact are usually characterized as 'hidden' in an intervention evaluation context, the purpose of the focus-group interviews will be to enable the identification of the unobservable mechanisms generating outcomes that statistical approaches do not elucidate [70]. The focus-group interviews will be held in continuation of the last GtB session in November 2024 to identify themes across the entire program whilst ensuring that the pupils are able to recall their experiences. Semi-structured interviews will be held with four to five garden facilitators and will take place at the end of the first half of the program period (after four sessions) as well as at the end of the program period (after eight sessions).

We will develop two separate interview guides for the facilitation of the focus-group interviews and semi-structured interviews. The theoretical underpinnings of GtB and theoretical frameworks of FL and CCL will guide the development of the guides. The interview guide for focus-group interviews with participating pupils will thus consist of both open-ended questions exploring the pupils' experiences with GtB, as well as evaluative questions to address specific components of GtB, which aimed to influence the pupils' FL and CCL. The interview guide for semi-structured interviews with garden facilitators will cover both open-ended questions focusing on the facilitators' general experience with the implementation of GtB activities, as well as their strategies and practices for achieving effective learning methods and components of GtB, including pedagogical and contextual factors contributing to these processes [73]. In total, two researchers will be present at each interview to moderate the discussions and take notes. All interviews will be recorded.

**2.6.2 Data analysis.** Data analysis will focus on the mechanisms operating at GtB which contribute to the promotion FL and CCL as well as to the successful implementation of GtB. However, the analysis will also apply an abductive approach, and thus remain open to emerging themes for further theory-building in relation to the program theory [44,74]. The focus-group interviews and semi-structured interviews will undergo full verbatim transcription and coding will be performed in the NVivo software. With an offset in a thematic analysis approach [75], a first-line coding will be guided by the theoretical frameworks and previous literature within the field, which will be supplemented by an abductive reasoning to the data. This will allow for the creation of coding matrixes and generation of patterns and recurring themes within the data. Data saturation will be determined when no new themes or a clear consistency start to emerge from the data and a full understanding of the conceptual categories has been acquired. The final pool of emerging themes will be discussed and agreed upon among the involved researchers. Finally, a triangulation of data across the focus-group interviews and semi-structured interviews will contribute to summarizing and verifying the emerging themes across participating pupils and garden facilitators [72].

## 3. Discussion

Previous research has indicated that school gardens can improve pupils' health and well-being [18,1]. The aim of the FoodACT study will be to investigate how the Danish GtB school garden program affects fourth and fifth grade pupils' FL, CCL, SM, and PA, as well as the implementation and mechanisms of impact of the program. This paper aims to present the study design, recruitment, data collection, measures, and analytical strategies that will be applied in the FoodACT study. The central strengths and limitations of these are discussed below.

Even though there are examples of individual schools running their own school gardens, the current practice of school gardens in Denmark is primarily led by the organization GtB in collaboration with municipalities [76]. Therefore, a central limitation of the study design will be that it is not feasible to randomize classes to an intervention and control group to ensure that these are comparable. RCTs are widely used in education research to reduce bias of estimates of intervention effects. However, this design also sets demands that can be hard to apply when investigating the impacts of initiatives that are already existing and thus implementable in real-life school settings. The FoodACT study will aim to investigate the impact of an already established school-garden program because this provided increased certainty about scalability and sustainable implementation. This gave the following barriers to using an RCT design: 1) The GtB program is implemented at the municipal level, which means that it is a political decision whether to financially prioritize participation in the school garden program

or not. It was therefore not financially feasible to randomize schools to the program. This, however, should not delimit the possibility of evaluating programs that are widely used in municipalities. On the contrary, their wide application provides a clear incentive for ensuring that their practice is evidence-based. 2) In other studies in Denmark, we have experienced several schools resisting to participate in RCT studies, especially when allocated to the control group [60,77]. Additionally, an RCT study of effects on accelerometer-measured PA with the proposed sample size as of this study would require greater financial support than received.

To accommodate this limitation, we will attempt to ensure that the intervention and control schools are comparable in terms of average disposable income at the municipal level. The GtB school garden program has existed through 18 years and their practical experience with organizing, recruiting, and teaching pupils in the school gardens will be a strength of this study, as it secures feasibility and an institutional structure for sustained program implementation. The garden sessions are led by GtB's own facilitators who hold a professional background in relevant fields, which ensures that the intervention is delivered similarly and with high quality across the different garden locations. The GtB program has previously been process evaluated using qualitative methods [78] and other researchers have investigated school subject integration within the GtB school gardens [79]. These findings have contributed to enhancing the program over the years, resulting in increased relevance for its target group.

The GtB program is chosen as the intervention case in the FoodACT study to evaluate the effects of a program that already takes place, and that is therefore feasible, implementable, and scalable in a real-world setting. The fact that the GtB program is already widely applied in Denmark highlights the importance of a thorough evaluation of its implementation, mechanisms and expected outcomes. Identifying the effects of interventions that already exist in a real-world setting can have a high societal impact and relevance (ecological validity) [47], which will be the main strength of this study. Furthermore, the use of validated instruments for measuring FL, SM and CCL, the inclusion of a control group, and the large sampling taking the clustered nature of data into account in the quasi-experimental studies, will contribute as main methodological strengths. Additionally, PA behavior will be collected using device-based measures with accelerometers, which have shown reliable estimates in children [80]. An additional strength of the study will be that it enables a comprehensive evaluation of school gardens through the use of both quantitative and qualitative methodologies on a broad range of outcomes related to children's health and health behaviors. Although the design, i.e., a natural experiment, does not make it possible to control what the pupils are doing where and when, as in more delimited controlled settings, the comprehensiveness of the FoodACT study has seldomly been applied in previous studies on the impacts of school gardens [1,18,22]. Thus, the study holds the potential to inspire future school garden research internationally, to inform schools and teachers to engage in practices related to health and sustainability, and to advise politicians in their process of prioritizing funds and pursuing both health and environmental considerations in all school policies [81].

Teachers will be present during the GtB activities and bear the responsibility of integrating garden-related tasks into the overarching curriculum. Additionally, the EOtC learning experiences possess the potential to impact students' social relations positively [15]. However, despite the significance of teacher roles and student social relations, our deliberate selection of outcome measures will diverge from these aspects. Instead, we will prioritize measures focusing on the critical contemporary challenges of health, well-being, and climate change. Furthermore, the qualitative component of SS3 will focus only on the outcome measures of FL and CCL, which may be viewed as a limitation with respect to the coherence between the various sub-studies, but which is, however, justified in the lack of focus on these concepts in school garden intervention evaluations [18].

## 4. Conclusion

The FoodACT study will contribute to a gap in the research literature by providing insights into the potential for school gardens to increase pupils' FL, CCL, SM, and PA behavior. The comprehensive study design will allow for a wide range of measures to be collected through both quantitative and qualitative approaches. By studying pupils' literacy in relation to food and climate change, motivation in school, and physical behaviors, the study will draw a comprehensive picture of the use of school gardens in promoting health but also addressing climate change in school lessons. Above all, the study will contribute to broadening the understanding of the potential benefits of school gardens in school health promotion strategies at the level of primary education. The results of the FoodACT study can thus be used as a practical example to the overall research in this area so that future policies can be guided by evidence as well as inform practice regarding school gardens on how to improve FL, CCL, SM and PA among pupils in primary schools.

## Supporting information

**S1 File. The FoodACT intervention TIDiR Checklist.**
(DOCX)

**S2 File. Spirit Checklist for The FoodACT Protocol.**
(DOCX)

**S3 File. Protokol_VEK_FoodACT_V1_withoutpics.**
(DOCX)

## Author contributions

**Conceptualization:** Anna Stage, Marie Caroline Vermund, Mads Bølling, Peter Elsborg.

**Investigation:** Anna Stage, Marie Caroline Vermund, Mads Bølling, Alberte Laura Oest Müllertz, Peter Elsborg.

**Methodology:** Anna Stage, Marie Caroline Vermund, Mads Bølling, Camilla Roed Otte, Glen Nielsen, Peter Elsborg.

**Project administration:** Anna Stage, Mads Bølling, Peter Bentsen, Glen Nielsen, Peter Elsborg.

**Resources:** Camilla Roed Otte.

**Supervision:** Mads Bølling, Camilla Roed Otte, Peter Bentsen, Glen Nielsen, Peter Elsborg.

**Visualization:** Mads Bølling.

**Writing – original draft:** Anna Stage, Marie Caroline Vermund, Glen Nielsen, Peter Elsborg.

**Writing – review & editing:** Anna Stage, Marie Caroline Vermund, Mads Bølling, Camilla Roed Otte, Alberte Laura Oest Müllertz, Peter Bentsen, Glen Nielsen, Peter Elsborg.

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
