## [Decision Letter · Decision Letter 0]

7 Jan 2025

PONE-D-24-43563The impact of a school garden program on children’s food literacy, climate change literacy, school motivation, and physical activity: A study protocolPLOS ONE

Dear Dr. Stage,

Thank you for submitting your manuscript to PLOS ONE. After careful consideration, we feel that it has merit but does not fully meet PLOS ONE’s publication criteria as it currently stands. Therefore, we invite you to submit a revised version of the manuscript that addresses the points raised during the review process. First, I want to say that I am grateful to you for your patience while waiting for this decision. You will recall that I got in touch to clarify opposed vs suggested reviewers some time ago; unfortunately, none of these, nor various other potential reviewers that I identified, were in a position to complete a review. In the end, in the interest of avoiding further delays, I completed my own review of the manuscript as well as the one independent peer review that I received (so I am Reviewer 2, below); this is in line with PLOS ONE policy. Both the other reviewer and I agree this protocol is publishable with revisions. These do not imply changes to the design of the study, but primarily request considerable further detail and clarity of the proposed analyses in order to minimise ambiguity in the conduct of that final analysis. For example, your current description of background variables to be included in the analysis models is quite vague and needs to be much more specific, otherwise it is quite possible that two equally justifiable decisions on how to include them would result in different conclusions from the study.

We look forward to receiving your revised manuscript.

Kind regards,

Jake Anders

Academic Editor

PLOS ONE

Journal requirements: When submitting your revision, we need you to address these additional requirements. 1. Please ensure that your manuscript meets PLOS ONE's style requirements, including those for file naming. The PLOS ONE style templates can be found at https://journals.plos.org/plosone/s/file?id=wjVg/PLOSOne_formatting_sample_main_body.pdf and https://journals.plos.org/plosone/s/file?id=ba62/PLOSOne_formatting_sample_title_authors_affiliations.pdf. 2. We note that the grant information you provided in the ‘Funding Information’ and ‘Financial Disclosure’ sections do not match.  When you resubmit, please ensure that you provide the correct grant numbers for the awards you received for your study in the ‘Funding Information’ section. 3. Thank you for stating the following in the Competing Interests section: [MCV and CRO are both employed with Haver til Maver / Gardens to Bellies ].  Please confirm that this does not alter your adherence to all PLOS ONE policies on sharing data and materials, by including the following statement: ""This does not alter our adherence to  PLOS ONE policies on sharing data and materials.” (as detailed online in our guide for authors http://journals.plos.org/plosone/s/competing-interests).  If there are restrictions on sharing of data and/or materials, please state these. Please note that we cannot proceed with consideration of your article until this information has been declared.  Please include your updated Competing Interests statement in your cover letter; we will change the online submission form on your behalf. 4. Your ethics statement should only appear in the Methods section of your manuscript. If your ethics statement is written in any section besides the Methods, please move it to the Methods section and delete it from any other section. Please ensure that your ethics statement is included in your manuscript, as the ethics statement entered into the online submission form will not be published alongside your manuscript.  5. Please include captions for your Supporting Information files at the end of your manuscript, and update any in-text citations to match accordingly. Please see our Supporting Information guidelines for more information: http://journals.plos.org/plosone/s/supporting-information. 

Reviewers' comments:

Reviewer's Responses to Questions

**Comments to the Author**

1. Does the manuscript provide a valid rationale for the proposed study, with clearly identified and justified research questions?

Reviewer #1: Partly

Reviewer #2: Yes

2. Is the protocol technically sound and planned in a manner that will lead to a meaningful outcome and allow testing the stated hypotheses?

Reviewer #1: Partly

Reviewer #2: Partly

3. Is the methodology feasible and described in sufficient detail to allow the work to be replicable?

Reviewer #1: Yes

Reviewer #2: No

4. Have the authors described where all data underlying the findings will be made available when the study is complete?

Reviewer #1: Yes

Reviewer #2: Yes

5. Is the manuscript presented in an intelligible fashion and written in standard English?

Reviewer #1: Yes

Reviewer #2: Yes

6. Review Comments to the Author

You may also provide optional suggestions and comments to authors that they might find helpful in planning their study.

Reviewer #1: The citation in the text is to be separated from the word e.g. performance(1–3) to be presented as performance (1-3). This applies throughout the manuscript.

Line 220-221: SS2 and SS3 to be spelled out as sub-study 2 and sub-study 3 or SS is to be denoted/indicated in Table 1.

Line 227-231: the detailed sampling process/procedure is to be described.

Line 244: the language version of the questionnaires/inventories/surveys/tools used in the study is to be clearly stated.

Line 276: the classification for SOSC is to be provided e.g. Class I to Class VI.

Line 284: define what is home economics.

Line 274-296: the classification of categories (categorical) data of these variables is to be provided e.g. Variable A 1= , 2= 3= , 4=

Similarly, for the outcome measures, how FL, CCL, and SM will be classified and analysed e.g. scale or categorical data. For categorical data, if possible the coding for the stems is to be included. This applies for other tools in Line 336-359.

Line 303-306: the sentence is not clear and requires revision. Need to state clearly how many classes and how many students in a class required for the sample size.

Line 305: the power is to be stated.

For SS1 and SS2: Were the outcome variables simultaneously or independently analysed?

Multiple testing correction is to be applied where required.

Line 319: SES and home economics may be collinear. If collinearity is present, it should be addressed accordingly.

Line 322: the statistical software, its version and publisher name is to be stated.

Line 343: typo ‘school garden sessions: Will’

Line 361: how the sampling is done to be clearly stated.

Line 364-365: the sentence is unclear. More information is to be provided.

Line 374: the reason for these variables and not others were controlled is to be stated.

Line 416: typo tour

Line 438: the version of the software is to be stated. NVIVO is to be written as NVivo.

Not all references conformed to the journal format.

Reviewer #2: * I would encourage you to provide a more nuanced discussion of the rationale for a quasi-experiment over an RCT in your context. It is fair to say that RCTs (often) prioritise internal over external validity, but establishing internal validity is important as a pre-condition for understanding external validity. I would encourage you to discuss why external validity is the priority for this particular intervention and study at this particular time, given the existing evidence base.

* Accepting, however, that a clustered quasi-experimental design is the feasible design of this study, the credibility of the results will be much stronger if it is clearly explained how the comparison schools were recruited to maximise the comparability across intervention and comparison groups. The current statement that “recruitment and inclusion of control classes will be based on similarity with the intervention schools regarding geographical proximity and average disposable income at the municipal level” is rather vague. What does it actually look like in practice?

* You set out sample sizes in lines 227-228 but it is not clear (at least at this point) why these particular sizes are chosen. They don’t seem to be based on power calculations (since these are reported later and use different, more conservative, values), so is it logistical constraints, or something?

* Background descriptive measures are extremely vaguely specified and, given these are proposed to be included in the model for estimating impact, should be much more explicit to ensure there is less ambiguity in that modelling. At the moment there would be huge variation in the models you could actually run to estimate the impact of the study, which may lead to differing conclusions and, hence, undermines the benefits of having pre-registered this study with this protocol.

* On the models themselves (sections 2.4.5 and 2.5.4), it would make the model you propose to run much clearer to write it down algebraically. This interplays with the above suggestion to be far more explicit about the other covariates that are being included in the model.

* You sensibly say that you will estimate ICCs but do not say how. Will you extract these directly from your main impact estimation model? Some methodological literature instead advocates use of a model without the various covariates to estimate the ICC, so your current plan is ambiguous.

Minor points

* Line 218: invented should be invited?

* Line 314: factor should be intercept?

* Line 321-322: “two-tailed tests with p-values below 0.05 will be considered statistically significant.” Is this meant to come earlier (~ line 316) to be about testing the coefficient on the focal interaction term.

* Line 361: in “containing of” the “of” is superfluous

* There are various other typos and the manuscript would benefit from a check to eliminate these, especially because some do result in ambiguity of meaning.

7. PLOS authors have the option to publish the peer review history of their article (what does this mean? ). If published, this will include your full peer review and any attached files.

**Do you want your identity to be public for this peer review?** For information about this choice, including consent withdrawal, please see our Privacy Policy .

Reviewer #1: No

Reviewer #2: **Yes: ** Jake Anders

---

## [Author Response · Author response to Decision Letter 1]

13 Feb 2025

Dear Reviewers,

Thank you for your valuable time and comments on our manuscript. We have carefully addressed each of your suggestions in the document "Response to Reviewers" and believe that these revisions have enhanced the quality of the manuscript.

Furthermore, we have thoroughly reviewed the journal’s submission requirements, as outlined in the decision letter. We have also double-checked the grant letter, both in the manuscript and on the online platform, and can confirm it is the correct version. Should you notice any discrepancies or errors, please do not hesitate to let us know.

Thank you once again for your constructive feedback.

Best Regards,

Anna Stage

---

## [Decision Letter · Decision Letter 1]

21 Feb 2025

The impact of a school garden program on children’s food literacy, climate change literacy, school motivation, and physical activity: A study protocol

PONE-D-24-43563R1

Dear Dr. Stage,

We’re pleased to inform you that your manuscript has been judged scientifically suitable for publication and will be formally accepted for publication once it meets all outstanding technical requirements.

Kind regards,

Jake Anders

Academic Editor

PLOS ONE

Additional Editor Comments (optional):

Reviewers' comments:

Reviewer's Responses to Questions

**Comments to the Author**

1. Does the manuscript provide a valid rationale for the proposed study, with clearly identified and justified research questions?

Reviewer #1: Partly

Reviewer #2: Yes

2. Is the protocol technically sound and planned in a manner that will lead to a meaningful outcome and allow testing the stated hypotheses?

Reviewer #1: Partly

Reviewer #2: Yes

3. Is the methodology feasible and described in sufficient detail to allow the work to be replicable?

Reviewer #1: Yes

Reviewer #2: Yes

4. Have the authors described where all data underlying the findings will be made available when the study is complete?

Reviewer #1: Yes

Reviewer #2: No

5. Is the manuscript presented in an intelligible fashion and written in standard English?

Reviewer #1: Yes

Reviewer #2: Yes

6. Review Comments to the Author

You may also provide optional suggestions and comments to authors that they might find helpful in planning their study.

Reviewer #1: The authors have addressed the comments.

No further comments.

The manuscript is acceptable for publication.

Reviewer #2: Thank you for thoroughly addressing the issues that I raised. I did slightly um and ah about the "Have the authors described where all data underlying the findings will be made available when the study is complete?" question, as your declaration doesn't say "where" but just that they will be made available — however, your declaration is very similar to the examples provided. As such, I would encourage you to provide further information on how/where you plan to make the data available, if this is possible at this point, but understand that it may not be clear yet.

7. PLOS authors have the option to publish the peer review history of their article (what does this mean? ). If published, this will include your full peer review and any attached files.

**Do you want your identity to be public for this peer review?** For information about this choice, including consent withdrawal, please see our Privacy Policy .

Reviewer #1: No

Reviewer #2: **Yes: ** Jake Anders

---

## [Editor Report · Acceptance letter]

PONE-D-24-43563R1

PLOS ONE

Dear Dr. Stage,

I'm pleased to inform you that your manuscript has been deemed suitable for publication in PLOS ONE. Congratulations! Your manuscript is now being handed over to our production team.

Kind regards,

on behalf of

Prof. Jake Anders

Academic Editor

PLOS ONE